# Differential Privacy of Dirichlet Posterior Sampling

## Abstract

We study the inherent privacy of releasing a single sample from a Dirichlet posterior distribution. As a complement to the previous study that provides general theories on the differential privacy of posterior sampling from exponential families, this study focuses specifically on the Dirichlet posterior sampling and its privacy guarantees. With the notion of truncated concentrated differential privacy (tCDP), we are able to derive a simple privacy guarantee of the Dirichlet posterior sampling, which effectively allows us to analyze its utility in various settings. Specifically, we provide accuracy guarantees of the Dirichlet posterior sampling in Multinomial-Dirichlet sampling and private normalized histogram publishing.

## 1 Introduction

The Bayesian framework provides a way to perform statistical analysis by combining prior beliefs with real-life evidence. At a high level, the belief and the evidence are assumed to be described by probabilistic models. As we receive new data, our belief is updated accordingly via the Bayes' theorem, resulting in the so-called posterior belief. The posterior tells us how much we are uncertain about the model's parameters.

The Dirichlet distribution is usually chosen as the prior when performing Bayesian analysis on discrete variables, as it is a conjugate prior to the categorical and multinomial distributions. Specifically, Dirichlet distributions are often used in discrete mixture models, where a Dirichlet prior is put on the mixture weights [LW92; MMR05]. Such models have applications in NLP [PB98], biophysical systems [Hin15], accident analysis [de 06], and genetics [BHW00; PM01; CWS03]. In all of these studies, samplings from Dirichlet posteriors arise when performing Markov chain Monte Carlo methods for approximate Bayesian inference.

Dirichlet posterior sampling also appears in other learning tasks. For example, in Bayesian active learning, it arises in Gibbs sampling, which is used to approximate the posterior of the classifier over the labeled sample [NLYCC13]. In Thompson sampling for multi-armed bandits, one repeatedly draws a sample from the Dirichlet posterior of each arm, and picks the arm whose sample maximizes the reward [ZHGSY20; AAFK20; NIK20]. And in Bayesian reinforcement learning, state-transition probabilities are sampled from the Dirichlet posterior over past observed states [Str00; ORR13].

Dirichlet posterior sampling can also be used for data synthesis. Suppose that we have a histogram $(x_1, \ldots, x_d)$ of actual data. An approximate discrete distribution of this histogram can be obtained by drawing a sample $\mathbf{Y}$ from $\text{Dirichlet}(x_1 + \alpha_1, \ldots, x_d + \alpha_d)$, where $\alpha_1, \ldots, \alpha_d$ are prior parameters. Then synthetic data is produced by repeatedly drawing from $\text{Multinomial}(\mathbf{Y})$. There are many studies on data synthesis that followed this approach [AV08; MKAGV08; RWZ14; PG14; SJGLY17].

In the above examples, the data that we integrate into these tasks might contain sensitive information. Thus it is important to ask: how much of the information is protected from the Dirichlet samplings? The goal of this study is to find an answer to this question.

The mathematical framework of differential privacy (DP) [DMNS06] allows us to quantify how much the privacy of the Dirichlet posterior sampling is affected by the prior parameters $\alpha_1, \ldots, \alpha_d$. In the definition of DP, the privacy of a randomized algorithm is measured by how much its distribution changes upon perturbing a single data point of the input. Nonetheless, this notion might be too strict for the Dirichlet distribution, as a small perturbation of a near-zero parameter can cause a large distribution shift. Thus, it might be more appropriate to rely on one of several relaxed notions of DP, such as approximate differential privacy, Rényi differential privacy, or concentrated differential privacy. It is natural to wonder if the Dirichlet posterior sampling satisfies any of these definitions.

## 1.1 Overview of Our results

This study focuses on the privacy and utility of Dirichlet posterior sampling. In summary, we provide a closed-form privacy guarantee of the Dirichlet posterior sampling, which in turn allows us to effectively analyze its utility in various settings.

**§3 Privacy.** We study the role of the prior parameters in the privacy of the Dirichlet posterior sampling. Theorem 1 is our main result, where we provide a guaranteed upper bound for truncated concentrated differential privacy (tCDP) of the Dirichlet posterior sampling. In addition, we convert the tCDP guarantee into an approximate differential privacy guarantee in Corollary 2.

**§4 Utility.** Using the tCDP guarantee, we investigate the utility of Dirichlet posterior sampling applied in two specific applications:

- In Section 4.1, we consider one-time sampling from a Multinomial-Dirichlet distribution. But instead of directly sampling from this distribution, we sample from another distribution with larger prior parameters. The accuracy is then measured by the KL-divergence between the original and the private distributions.

- In Section 4.2, we use the Dirichlet posterior sampling for a private release of a normalized histogram. In this case, the accuracy is measured by the mean-squared error between the sample and the original normalized histogram.

In both tasks, we compute the sample size that guarantees the desired level of accuracy. In the case of private histogram publishing, we also compare the Dirichlet posterior sampling to the Gaussian mechanism.

## 1.2 Related work

There are several studies on the differential privacy of posterior sampling. Wang, Fienberg, and Smola [WFS15] showed that any posterior sampling with the log-likelihood bounded by $B$ is $4B$-differentially private. However, the likelihoods that we study are not bounded away from zero; they have the form $\prod_i p_i^{x_i}$ which becomes small when one of the $p_i$'s is close to zero. Dimitrakakis, Nelson, Zhang, Mitrokotsa, and Rubinstein [DNZMR17] showed that if the condition on the log-likelihood is relaxed to the Lipschitz continuity with high probability, then one can obtain the approximate DP. Nonetheless, with the Dirichlet density, it is difficult to compute the probability of events in which the Lipschitz condition is satisfied.

In the case that the sufficient statistics $\mathbf{x}$ has finite $\ell^1$-sensitivity, Foulds, Geumlek, Welling and Chaudhuri [FGWC16] suggested adding Laplace noises to $\mathbf{x}$. Suppose that $\mathbf{y}$ is the output; they showed that sampling from $p(\theta|\mathbf{y})$ is differentially private and as asymptotically efficient as sampling from $p(\theta|\mathbf{x})$. However, for a small sample size, the posterior over the noisy statistics might be too far away from the actual posterior. Bernstein and Sheldon [BS18] thus proposed to approximate the joint distribution $p(\theta, \mathbf{x}, \mathbf{y})$ using Gibbs sampling, which is then integrated over $\mathbf{x}$ to obtain a more accurate posterior over $\mathbf{y}$.

Geumlek, Song, and Chaudhuri [GSC17] were the first to study the posterior sampling with the RDP. Even though they provided a general framework to find $(\lambda, \epsilon)$-RDP guarantees for exponential families, explicit forms of $\epsilon$ and the upper bound of $\lambda$ were not given. In contrast, our tCDP guarantees of the Dirichlet posterior sampling imply an explicit expression for $\epsilon$, and also an upper bound for $\lambda$.

The privacy of data synthesis via sampling from $\mathrm{Multinomial}(\mathbf{Y})$, where $\mathbf{Y}$ is a discrete distribution drawn from the Dirichlet posterior, was first studied by Machanavajjhala, Kifer, Abowd, Gehrke, and Vilhuber [MKAGV08]. They showed that the data synthesis is $(\varepsilon,\delta)$-*probabilistic* DP, which implies $(\varepsilon,\delta)$-approximate DP. However, as their privacy analysis includes the sampling from $\mathrm{Multinomial}(\mathbf{Y})$, their privacy guarantee depends on the number of synthetic samples. In contrast, we show that the one-time sampling from the Dirichlet posterior is approximate DP, which by the post-processing property allows us to sample from $\mathrm{Multinomial}(\mathbf{Y})$ as many times as we want while retaining the same privacy guarantee.

The Dirichlet mechanism was first introduced by Gohari, Wu, Hawkins, Hale, and Topcu [GWHHT21]. Originally, the Dirichlet mechanism takes a discrete distribution $\mathbf{p} := (p_1, \ldots, p_d)$ and draws one sample $\mathbf{Y} \sim \mathrm{Dirichlet}(rp_1, \ldots, rp_d)$. Note the absence of the prior parameters, which makes $\mathbf{Y}$ an unbiased estimator of $\mathbf{p}$. But this comes with a cost, as the worst case of privacy violation occurs when almost all of the parameters are close to zero. The authors avoided this issue by restricting the input space to a subset of the unit simplex, with some of the $p_i$'s bounded below by a fixed positive constant. This results in complicated expressions for the privacy guarantees as they involve a minimization problem over the restricted domain. In this study, we take a different approach by adding prior parameters to the Dirichlet mechanism. As a result, we obtain a biased algorithm that requires no assumption on the input space and has simpler forms of privacy guarantees.

## 1.3 Notations

We let $\mathbb{R}^d_{\geq 0}$ be the set of $d$-tuples of non-negative real numbers and $\mathbb{R}^d_{>0}$ be the set of $d$-tuples of positive real numbers. We assume that all vectors are $d$-dimensional where $d \geq 2$. The notations for all vectors are always in bold. Specifically, $\mathbf{x} := (x_1, \ldots, x_d) \in \mathbb{R}^d_{\geq 0}$ consists of sample statistics of the data and $\boldsymbol{\alpha} := (\alpha_1, \ldots, \alpha_d) \in \mathbb{R}^d_{>0}$ consists of the prior parameters. The vector $\mathbf{p} := (p_1, \ldots, p_d)$ always satisfies $\sum_i p_0 = 1$. The number of observations is always $N$. We also denote $x_0 := \sum_i x_i$ and $\alpha_0 := \sum_i \alpha_i$. For any vectors $\mathbf{x}, \mathbf{x}'$ and scalar $r > 0$, we write $\mathbf{x} + \mathbf{x}' := (x_1 + x'_1, \ldots, x_d + x'_d)$ and $r\mathbf{x} := (rx_1, \ldots, rx_d)$. For any positive reals $x$ and $x'$, the notation $x \propto x'$ means $x = Cx'$ for some constant $C > 0$, $x \approx x'$ means $cx' \leq x \leq Cx'$ for some $c, C > 0$, and $x \lesssim x'$ means $x \leq Cx'$ for some $C > 0$. Lastly, $\|\mathbf{x}\|_\infty := \max_i |x_i|$ is the $\ell^\infty$ norm of $\mathbf{x}$.

# 2 Background

## 2.1 Privacy models

**Definition 2.1** (Pure and Approximate DP [DMNS06])**.** A randomized mechanism $M : \mathcal{X}^n \to \mathcal{Y}$ is $(\varepsilon, \delta)$-differentially private $((\varepsilon, \delta)$-DP) if for any datasets $x, x'$ differing on a single entry, and all events $E \subset \mathcal{Y}$,
$$\mathbb{P}[M(x) \in E] \leq e^\varepsilon \mathbb{P}[M(x') \in E] + \delta.$$
If $M$ is $(\varepsilon, 0)$-DP, then we say that it is $\varepsilon$-differential privacy ($\varepsilon$-DP).

The term *pure differential privacy* (pure DP) refers to $\epsilon$-differential privacy, while *approximate differential privacy* (approximate DP) refers to $(\varepsilon, \delta)$-DP when $\delta > 0$.

In contrast to pure and approximate DP, the next definitions of differential privacy are defined in terms of the Rényi divergence between $M(x)$ and $M(x')$:

**Definition 2.2** (Rényi Divergence [Rén61])**.** Let $P$ and $Q$ be probability distributions. For $\lambda \in (1, \infty)$ the Rényi divergence of order $\lambda$ between $P$ and $Q$ is defined as
$$\mathrm{D}_\lambda(P\|Q) := \frac{1}{\lambda - 1} \log \int P(y)^\lambda Q(y)^{1-\lambda} \, dy = \frac{1}{\lambda - 1} \log\left( \mathop{\mathbb{E}}_{y \sim P}\left[ \frac{P(y)^{\lambda-1}}{Q(y)^{\lambda-1}} \right]. \right)$$

**Definition 2.3** (tCDP and zCDP [BDRS18; BS16]). A randomized mechanism $M : \mathcal{X}^n \to \mathcal{Y}$ is $\omega$-truncated $\rho$-concentrated differentially private (($\rho, \omega$)-tCDP) if for any datasets $x, x'$ differing on a single entry and for all $\lambda \in (1, \omega)$,

$$\mathrm{D}_\lambda(M(x)\|M(x')) \leq \lambda\rho.$$

If $M$ is $(\rho, \infty)$-tCDP, then we say that it is $\rho$-zero-concentrated differential privacy ($\rho$-zCDP).

Note that both tCDP and zCDP have the composition and post-processing properties. Intuitively, $\rho$ controls the expectation and standard deviation of the privacy loss random variable: $Z = \log \frac{P[M(x)=Y]}{P[M(x')=Y]}$, where $Y$ has density $M(x)$, and $\omega$ controls the number of standard deviations for which $Z$ concentrates like a Gaussian. A smaller $\rho$ and larger $\omega$ correspond to a stronger privacy guarantee. It turns out that tCDP implies approximate DP:

**Lemma 1** (From tCDP to Approximate DP [BDRS18]). *Let $\delta > 0$. If $M$ is a $(\rho, \omega)$-tCDP mechanism, then it also satisfies $(\varepsilon, \delta)$-DP with*

$$\varepsilon = \begin{cases} \rho + 2\sqrt{\rho\log(1/\delta)} & \text{if } \log(1/\delta) \leq (\omega-1)^2\rho \\ \rho\omega + \frac{\log(1/\delta)}{\omega-1} & \text{if } \log(1/\delta) > (\omega-1)^2\rho \end{cases}.$$

## 2.2 Dirichlet distribution

For $\boldsymbol{\alpha} \in \mathbb{R}_{>0}^d$, the Dirichlet distribution $\mathrm{Dirichlet}(\boldsymbol{\alpha})$ is a continuous distribution of $d$-dimensional probability vectors i.e. vectors whose coordinate sum is equal to 1. The density function of $\mathbf{Y} \sim \mathrm{Dirichlet}(\boldsymbol{\alpha})$ is given by:

$$p(\mathbf{y}) = \frac{1}{B(\boldsymbol{\alpha})} \prod_{i=1}^{d} y_i^{\alpha_i - 1},$$

where $B(\boldsymbol{\alpha})$ is the *beta function*, which can be written in terms of the gamma function:

$$B(\boldsymbol{\alpha}) = \frac{\prod_i \Gamma(\alpha_i)}{\Gamma(\sum_i \alpha_i)}. \tag{1}$$

## 2.3 Dirichlet posterior sampling

We consider the prior $\mathrm{Dirichlet}(\boldsymbol{\alpha})$ and the likelihood of the form $p(\mathbf{x}|\mathbf{y}) \propto \prod_{i=1}^{d} y_i^{x_i}$ where $\mathbf{x} \in \mathbb{R}_{\geq 0}^d$ consists of sample statistics of the dataset. The *Dirichlet posterior sampling* is a one-time sampling:

$$\mathbf{Y} \sim \mathrm{Dirichlet}(\mathbf{x} + \boldsymbol{\alpha}).$$

There is a modification of the sampling which introduces a concentration parameter $r > 0$, and instead we sample from $\mathrm{Dirichlet}(r\mathbf{x} + \boldsymbol{\alpha})$ [GSC17; GWHHT21]. Smaller values of $r$ make the sampling more private, and larger values of $r$ make $\mathbf{Y}$ a closer approximation of $\mathbf{x}$. Even though the case $r = 1$ is the main focus of this study, our main privacy results can be easily extended to other values of $r$ as we will see at the end of Section 3.1.

Consider a special case where $\mathbf{x} = \mathbf{p}$ is an empirical distribution derived from the dataset, and we want $\mathbf{Y}$ to be a private approximation of $\mathbf{p}$; the sampling $\mathbf{Y} \sim \mathrm{Dirichlet}(r\mathbf{p} + \boldsymbol{\alpha})$ is called the *Dirichlet mechanism* [GWHHT21]. It is interesting to note that the Dirichlet mechanism is a form of the exponential mechanism [MT07]: let $r > 0$ be the privacy parameter, $\mathrm{Dirichlet}(\boldsymbol{\alpha})$ be the prior, and the negative KL-divergence be the score function of the exponential mechanism. Then the output $\mathbf{Y}$ of this mechanism is distributed according to the following density function:

$$\frac{\exp(-r\,\mathrm{D_{KL}}(\mathbf{p}, \mathbf{y})) \prod_i y_i^{\alpha_i - 1}}{\int \exp(-r\,\mathrm{D_{KL}}(\mathbf{p}, \mathbf{y})) \prod_i y_i^{\alpha_i - 1} d\mathbf{y}} \propto \exp\left( r \sum_{i, p_i \neq 0} p_i \log(y_i/p_i) \right) \prod_i y_i^{\alpha_i - 1}$$

$$\propto \prod_{i, p_i \neq 0} y_i^{rp_i} \prod_i y_i^{\alpha_i - 1} = \prod_i y_i^{rp_i + \alpha_i - 1},$$

which is exactly the density function of $\mathrm{Dirichlet}(r\mathbf{p} + \boldsymbol{\alpha})$.

## 2.4 Polygamma functions

In most of this study, we take advantage of several nice properties of the log-gamma function and its derivatives. Specifically, $\psi(x) := \frac{d}{dx} \log \Gamma(x)$ is concave and increasing, while its derivative $\psi'(x)$ is positive, convex, and decreasing. In addition, $\psi'$ can be approximated by the reciprocals:

$$\frac{1}{x} + \frac{1}{2x^2} < \psi'(x) < \frac{1}{x} + \frac{1}{x^2},$$

which implies that $\psi'(x) \approx \frac{1}{x^2}$ as $x \to 0$ and $\psi'(x) \approx \frac{1}{x}$ as $x \to \infty$.

# 3 Main privacy results

## 3.1 Truncated concentrated differential privacy

**Theorem 1.** *Let* $\boldsymbol{\alpha} \in \mathbb{R}^d_{>0}$ *and* $\alpha_m := \min_i \alpha_i$. *Let* $\gamma \in (0, \alpha_m)$. *Let* $\Delta_2, \Delta_\infty > 0$ *be constants that satisfy* $\sum_i (x_i - x'_i)^2 \leq \Delta_2^2$ *and* $\max_i |x_i - x'_i| \leq \Delta_\infty$ *whenever* $\mathbf{x}, \mathbf{x}' \in \mathbb{R}^2_{\geq 0}$ *are sample statistics of any two datasets differing on a single entry. The one-time sampling from* $\mathrm{Dirichlet}(\mathbf{x} + \boldsymbol{\alpha})$ *is* $(\rho, \omega)$-*tCDP, where* $\omega = \frac{\gamma}{\Delta_\infty} + 1$ *and*

$$\rho = \frac{1}{2} \Delta_2^2 \psi'(\alpha_m - \gamma). \tag{2}$$

Note that $(\rho, \infty)$-tCDP is not obtainable, as the ratio between two Dirichlet densities blows up as $\omega \to \infty$. We present here a short proof that skips some calculations (see Appendix 1 for a full proof).

*proof.* Consider any $\lambda \in \left(1, \frac{\gamma}{\Delta_\infty} + 1\right)$. Let $\mathbf{u} := \mathbf{x} + \boldsymbol{\alpha}$ and $\mathbf{u}' := \mathbf{x}' + \boldsymbol{\alpha}'$. Let $P(\mathbf{y})$ be the density of $\mathrm{Dirichlet}(\mathbf{u})$ and $P'(\mathbf{y})$ be the density of $\mathrm{Dirichlet}(\mathbf{u}')$. A quick calculation shows that:

$$\mathbb{E}_{\mathbf{y} \sim P(\mathbf{y})} \left[ \frac{P(\mathbf{y})^{\lambda-1}}{P'(\mathbf{y})^{\lambda-1}} \right] = \frac{B(\mathbf{u}')^{\lambda-1}}{B(\mathbf{u})^{\lambda-1}} \cdot \frac{B(\mathbf{u} + (\lambda-1)(\mathbf{u} - \mathbf{u}'))}{B(\mathbf{u})}. \tag{3}$$

We take the logarithm on both sides and apply the second-order Taylor expansion to the following $G(u_i, u'_i)$ and $H(u_i, u'_i)$ terms that appear on the right-hand side. As a result, there exist $\xi$ between $u_i + (\lambda-1)(u_i - u'_i)$ and $u_i$, and $\xi'$ between $u_i$ and $u'_i$ such that

$$G(u_i, u'_i) := (\lambda-1)(\log \Gamma(u'_i) - \log \Gamma(u_i))$$

$$= -(\lambda-1)(x_i - x'_i)\psi(u_i) + \frac{1}{2}(\lambda-1)(x_i - x'_i)^2 \psi'(\xi') \tag{4}$$

$$H(u_i, u'_i) := \log \Gamma(u_i + (\lambda-1)(u_i - u'_i)) - \log \Gamma(u_i)$$

$$= (\lambda-1)(x_i - x'_i)\psi(u_i) + \frac{1}{2}(\lambda-1)^2(x_i - x'_i)^2 \psi'(\xi), \tag{5}$$

Note that $\psi'$ is increasing. If $x_i > x'_i$, then $\xi$ and $\xi'$ are bounded below by $u'_i \geq \alpha_m$. On the other hand, if $x_i \leq x'_i$, then $\xi$ and $\xi'$ are bounded below by $u_i - (\lambda-1)|u_i - u'_i|$. The condition $\lambda < \frac{\gamma}{\Delta_\infty} + 1$ guarantees that $u_i - (\lambda-1)|u_i - u'_i| > \alpha_m - \gamma$. All cases considered, we have

$$G(u_i, u'_i) + H(u_i, u'_i) \leq \frac{1}{2}\big((\lambda-1) + (\lambda-1)^2\big)(x_i - x'_i)^2 \psi'(\alpha_m - \gamma)$$

$$= \frac{1}{2}\lambda(\lambda-1)(x_i - x'_i)^2 \psi'(\alpha_m - \gamma).$$

Denoting $u_0 := \sum_i u_i$ and $u'_0 := \sum_i u'_i$, the same argument shows that $G(u_0, u'_0) + H(u_0, u'_0) > 0$. Therefore,

$$D_\lambda(P(\mathbf{y}) \| P'(\mathbf{y})) = \frac{1}{\lambda-1} \left( \sum_i (G(u_i, u'_i) + H(u_i, u'_i)) - G(u_0, u'_0) - H(u_0, u'_0) \right)$$

$$< \frac{1}{\lambda-1} \sum_i (G(u_i, u'_i) + H(u_i, u'_i))$$

$$\leq \frac{1}{2}\lambda \sum_i (x_i - x'_i)^2 \psi'(\alpha_m - \gamma) \leq \frac{1}{2}\lambda \Delta_2^2 \psi'(\alpha_m - \gamma). \qquad \square$$

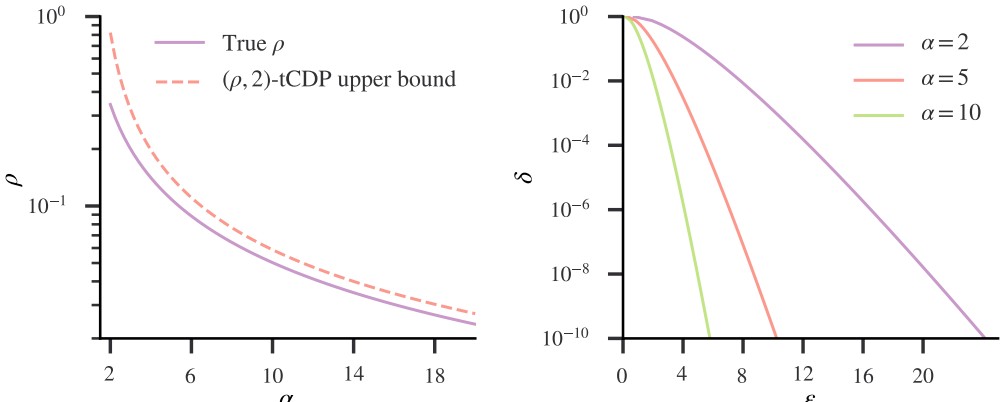

Figure 1: Left: the actual values of $\rho = \frac{1}{2} D_2(P\|P')$ and the worst case $(\rho, 2)$-tCDP guarantees (2) at $\Delta_2^2 = \Delta_\infty = 1$. Here, $P$ and $P'$ are Dirichlet posterior densities over $\mathbf{x} = (11, 8, 65, 25, 38, 0)$, $\mathbf{x}' = (11, 8, 65, 25, 38, 1)$, and $\boldsymbol{\alpha} = (\alpha, \dots, \alpha)$. Right: comparison between $(\varepsilon, \delta)$-DP guarantees of the Dirichlet posterior samplings (8) with different uniform priors: $\boldsymbol{\alpha} = (\alpha, \dots, \alpha)$.

The guaranteed upper bound (2) is independent of the sample statistics. As a result, the bound applies even in worst settings i.e., when $x_i = 0$ and $x_i' = \Delta_\infty$, or vice versa, for some $i$. As we can see in Figure 1, the upper bound is a close approximation to the actual value of $\rho$ when $x_6 = 0$ and $x_6' = 1$. However, being a sample independent bound, the difference becomes substantial when all $x_i$'s are large. There is one way to get around this issue: if there is no privacy violation in assuming that the sample statistics are always bounded below by some threshold $\tau$, then we can incorporate the threshold into the prior (thus $\psi'(\alpha_m - \gamma)$ in (2) is replaced by $\psi'(\alpha_m + \tau - \gamma)$).

The parameter $\gamma$ allows us to adjust the moment bound $\omega$ as desired. Even though a higher $\omega$ usually leads to a better privacy guarantee, there are two downsides to picking $\gamma$ close to $\alpha_m$ in this case. First, note that $\rho$ contains $\psi'(\alpha_m - \gamma)$; as $\gamma \to \alpha_m$, the value of $\rho$ diverges to $\infty$, leading to a weaker privacy guarantee instead. Second, as the Taylor approximation (5) is accurate when $u_i$ is close to $u_i + (\lambda - 1)(u_i - u_i')$, having a large value of $\lambda$ would push the guaranteed upper bound away from the actual privacy loss. Thus it is recommended to pick $\gamma$ so that $\gamma/\Delta_\infty \geq 1$ and $\alpha_m - \gamma \gg 0$. Alternatively, we can choose the value of $\gamma$ that minimizes $\varepsilon$ when converting from tCDP to $(\varepsilon, \delta)$-DP using Lemma 1—this method will be explored in the next subsection.

Theorem 1 can be easily applied to sampling from $\text{Dirichlet}(r\mathbf{x} + \boldsymbol{\alpha})$. Replacing $\mathbf{x}$ with $r\mathbf{x}$, we have $\Delta_2$ replaced by $r\Delta_2$ and $\Delta_\infty$ replaced by $r\Delta_\infty$. Consequently, the sampling is $\left(\rho, \frac{\gamma}{r\Delta_\infty} + 1\right)$-tCDP, where $\rho = \frac{1}{2} r^2 \Delta_2^2 \psi'(\alpha_m - \gamma)$. In Appendix 4, we analyze the scaling of $r$ in conjunction with $\alpha_m$ at a fixed privacy budget $\rho$.

## 3.2 Approximate differential privacy

We now convert the tCDP guarantee to an approximate DP guarantee. Let $\delta \in (0, 1)$. Using Lemma 1, the Dirichlet posterior sampling with $\text{Dirichlet}(\boldsymbol{\alpha})$ as the prior is $(\varepsilon, \delta)$-DP with

$$\varepsilon = \begin{cases} \rho(\gamma) + 2\sqrt{\rho(\gamma)\log(1/\delta)} & \text{if } \log(1/\delta) \leq \gamma^2\rho(\gamma)/\Delta_\infty^2 \\ \rho(\gamma)\left(\frac{\gamma}{\Delta_\infty} + 1\right) + \frac{\log(1/\delta)\Delta_\infty}{\gamma} & \text{if } \log(1/\delta) > \gamma^2\rho(\gamma)/\Delta_\infty^2 \end{cases}, \tag{6}$$

where $\rho(\gamma) = \frac{1}{2}\Delta_2^2\psi'(\alpha_m - \gamma)$.

We try to minimize $\epsilon$ by adjusting the value of $\gamma$. First, we consider the case $\log(1/\delta) \leq \gamma^2\rho(\gamma)/\Delta_\infty^2$. Since $\rho(\gamma)$ is a strictly increasing function of $\gamma$, both $\rho(\gamma) + 2\sqrt{\rho(\gamma)\log(1/\delta)}$ and $\gamma^2\rho(\gamma)/\Delta_\infty^2$ are both strictly increasing function of $\gamma$. Therefore, $\varepsilon$ is minimized at the minimum possible value of $\gamma$ in this case, that is, at the unique $\gamma_M$ that satisfies $\log(1/\delta) = \gamma_M^2\rho(\gamma_M)/\Delta_\infty^2 = \frac{1}{2}\gamma_M^2\Delta_2^2\psi'(\alpha_m - \gamma_M)/\Delta_\infty^2$.

Now we consider the second case, when $\gamma < \gamma_M$. As $\rho(\gamma)$ is an increasing positive convex function of $\gamma$, the function

$$f(\gamma) := \frac{1}{2}\Delta_2^2 \psi'(\alpha_m - \gamma)\left(\frac{\gamma}{\Delta_\infty} + 1\right) + \frac{\log(1/\delta)\Delta_\infty}{\gamma}; \qquad \gamma \in (0, \gamma_M], \tag{7}$$

is also convex in $\gamma$, and thus has a unique minimizer $\gamma_m \in (0, \gamma_M]$. Comparing to the first case, we have $f(\gamma_m) \leq f(\gamma_M) = \rho(\gamma_M) + 2\sqrt{\rho(\gamma_M)\log(1/\delta)}$. We then conclude that $\varepsilon = f(\gamma_m)$.

**Theorem 2.** *Let $\boldsymbol{\alpha} \in \mathbb{R}_{>0}^2$ and denote $\alpha_m = \min_i \alpha_i$. Let $\Delta_2, \Delta_\infty > 0$ be constants that satisfy $\sum_i (x_i - x_i')^2 \leq \Delta_2^2$ and $\max_i |x_i - x_i'| \leq \Delta_\infty$ whenever $\mathbf{x}, \mathbf{x}' \in \mathbb{R}_{\geq 0}^d$ are sample statistics of any two datasets differing on a single entry. For any $\delta \in (0,1)$, let $\gamma_M$ be the solution to the equation $\log(1/\delta) = \frac{1}{2}\gamma^2 \Delta_2^2 \psi'(\alpha_m - \gamma)/\Delta_\infty^2$. The one-time sampling from $\mathrm{Dirichlet}(\mathbf{x} + \boldsymbol{\alpha})$ is $(\varepsilon, \delta)$-DP, where*

$$\varepsilon = \min_{\gamma \in (0, \gamma_M]} f(\gamma). \tag{8}$$

Figure 1 shows how $\delta$ decays as a function of $\varepsilon$ at three different values of $\alpha_m$.

# 4 Utility

Using the results from the previous section, we analyze the Dirichlet posterior sampling's utility in two specific tasks.

## 4.1 Multinomial-Dirichlet sampling

Suppose that we are observing $N$ trials, each of which has $d$ possible outcomes. For each $i \in \{1, \ldots, d\}$, let $x_i$ be the number of times the $i$-th outcome was observed. Then we have the multinomial likelihood $p(\mathbf{x}|\mathbf{y}) \propto \prod_i y_i^{x_i}$. From this, we sample from the Dirichlet posterior:

$$\mathbf{Y} \sim \mathrm{Dirichlet}(\mathbf{x} + \boldsymbol{\alpha}). \tag{9}$$

Suppose that we want to sample from a true distribution $P_{\mathbf{X}} \sim \mathrm{Dirichlet}(\mathbf{x} + \boldsymbol{\alpha})$, but for privacy reasons, we instead sample from $Q_{\mathbf{x}} \sim \mathrm{Dirichlet}(\mathbf{x} + \boldsymbol{\alpha}')$ where $\alpha_i' > \alpha_i$ for all $i$. The utility of the privacy scheme is then measured by the KL-divergence between $P_{\mathbf{x}}$ and $Q_{\mathbf{x}}$. Assuming that $\mathbf{x}$ is an observation of $\mathrm{Multinomial}(\mathbf{p})$, the following Theorem tells us that, on average, the KL-divergence is small when the sample size is large, and the $p_i$'s are evenly distributed.

**Theorem 3.** *Let $\mathbf{p} := (p_1, \ldots, p_d)$ where $p_i > 0$ for all $i$ and $\sum_i p_i = 1$. Define a random variable $\mathbf{X} \sim \mathrm{Multinomial}(\mathbf{p})$. Let $P_{\mathbf{X}} \sim \mathrm{Dirichlet}(\mathbf{X} + \boldsymbol{\alpha})$ and $Q_{\mathbf{X}} \sim \mathrm{Dirichlet}(\mathbf{X} + \boldsymbol{\alpha}')$ where $\alpha_i' \geq \alpha_i \geq 1$ for all $i$. The following estimate holds:*

$$\mathbb{E}_{\mathbf{X}}[\mathrm{D}_{\mathrm{KL}}(P_{\mathbf{X}} \| Q_{\mathbf{X}})] \leq \frac{1}{N+1}\sum_i (\alpha_i' - \alpha_i)^2 \cdot \frac{1}{p_i}. \tag{10}$$

The proof is given in Appendix 2. Let us consider a simple privacy scheme where we fix $s > 0$ and let $\alpha_i' = \alpha_i + s$ for all $i$. Thus (10) becomes:

$$\mathbb{E}_{\mathbf{X}}[\mathrm{D}_{\mathrm{KL}}(P_{\mathbf{X}} \| Q_{\mathbf{X}})] \leq \frac{G(\mathbf{p})s^2}{N+1}, \tag{11}$$

where $G(\mathbf{p}) := \sum_i 1/p_i$. Now we take into account the privacy parameters. Let $\rho = \Delta_2^2 \psi'(\alpha_m - \gamma)$ and $\rho' = \Delta_2^2 \psi'(\alpha_m' - \gamma)$, where $\alpha_m = \min_i \alpha_i$, $\alpha_m' = \min_i \alpha_i'$, and $\gamma < \alpha_m$. Here, we approximate the values of $\psi'(\alpha_m - \gamma)$ and $\psi'(\alpha_m' - \gamma)$ under two regimes:

**High-privacy regime:** $\alpha_m' - \gamma > 1$. We have $\psi'(\alpha_m' - \gamma) \approx 1/(\alpha_m' - \gamma)$, which implies $\alpha_m' - \gamma \approx \Delta_2^2/\rho'$. We also have $\alpha_m - \gamma \approx \Delta_2^2/\rho$ for $\alpha_m - \gamma \geq 1$ and $\alpha_m - \gamma > (\alpha_m - \gamma)^2 \approx \Delta_2^2/\rho$ for $\alpha - \gamma < 1$. Thus we have the following bound for the right-hand side of (11):

$$\frac{G(\mathbf{p})s^2}{N+1} = \frac{G(\mathbf{p})(\alpha_m' - \alpha_m)^2}{N+1} \lesssim \frac{\Delta_2^4 G(\mathbf{p})}{N+1}\left(\frac{1}{\rho'} - \frac{1}{\rho}\right)^2 < \frac{\Delta_2^4 G(\mathbf{p})}{\rho'^2(N+1)}. \tag{12}$$

Consequently, we have $\mathrm{D}_{\mathrm{KL}}(P \| Q) < \epsilon$ for $N = \Omega\left(\frac{\Delta_2^4 G(\mathbf{p})}{\rho'^2 \epsilon}\right)$.

**Low-privacy regime:** $1 > \alpha'_m - \gamma > 0$. This is similar as above, except we have $\alpha'_m - \gamma \approx \Delta_2/\rho'^{1/2}$ and $\alpha_m - \gamma \approx \Delta_2/\rho^{1/2}$. Similar computation as (12) shows that $\mathrm{D}_{\mathrm{KL}}(P\|Q) < \epsilon$ when $N = \Omega\left(\frac{\Delta_2^2 G(\mathbf{p})}{\rho' \epsilon}\right)$.

We observe that, in both regimes, the sample size scales faster with respect to $\epsilon$ with a higher value of $G(\mathbf{p})$, which is associated with a higher number of outcomes $d$, and more concentrated multinomial parameter $\mathbf{p}$; this agrees with the result of our simulation in Appendix 3. Moreover, for small $\rho'$ the sample size scales as $1/\rho'^2$, while for large $\rho'$ the sample size scales as $1/\rho'$.

## 4.2 Private normalized histograms

Let $\mathbf{x} = (x_1, \ldots, x_d)$ be a histogram of $N$ observations and $\mathbf{p} := \mathbf{x}/N$. We can privatize $\mathbf{p}$ by sampling a probability vector: $\mathbf{Y} \sim \mathrm{Dirichlet}(\mathbf{x} + \boldsymbol{\alpha})$. Note that $\mathbf{Y}$ is a biased estimator of $\mathbf{p}$. Denoting $\alpha_0 := \sum_i \alpha_i$, the bias of each component of $\mathbf{Y}$ is given by $\mathbb{E}[\mathbf{Y}] - p_i$. Hence,

$$|\mathrm{Bias}(Y_i)| = \left| \frac{x_i + \alpha_i}{N + \alpha_0} - p_i \right| = \frac{|x_i \alpha_0 - N\alpha_i|}{N(N + \alpha_0)} \leq \frac{N\alpha_0}{N(N + \alpha_0)} = \frac{\alpha_0}{N + \alpha_0}.$$

Since $Y_i \sim \mathrm{Beta}(x_i + \alpha_i, N + \alpha_0 - x_i - \alpha_i)$ is $\frac{1}{4(N+\alpha_0+1)}$-sub-Gaussian [MA17], we have,

$$\mathbb{P}[|Y_i - p_i| > t + |\mathrm{Bias}(Y_i)|] \leq \mathbb{P}[|Y_i - \mathbb{E}[Y_i]| + |\mathrm{Bias}(Y_i)| > t + |\mathrm{Bias}(Y_i)|]$$
$$= \mathbb{P}[|Y_i - \mathbb{E}[Y_i]| > t]$$
$$\leq 2e^{-2t^2(N+\alpha_0+1)}.$$

With the union bound, we plug in $t = \sqrt{\frac{\log(2d/\beta)}{2(N+\alpha_0+1)}}$, for any $\beta \in (0, 1)$, to obtain the following accuracy guarantee of the private normalized histogram:

**Theorem 4.** *Let* $\mathbf{Y} \sim \mathrm{Dirichlet}(\mathbf{x} + \boldsymbol{\alpha})$, *where* $\mathbf{x} \in \mathbb{R}^d_{\geq 0}$ *and* $\boldsymbol{\alpha} \in \mathbb{R}^d_{>0}$, *and* $\mathbf{p} := \mathbf{x}/N$. *For any* $\beta \in (0,1)$, *with probability at least* $1 - \beta$, *the following inequality holds:*

$$\|\mathbf{Y} - \mathbf{p}\|_\infty \leq \sqrt{\frac{\log(2d/\beta)}{2(N + \alpha_0 + 1)}} + \frac{\alpha_0}{N + \alpha_0}. \tag{13}$$

Given $\epsilon > 0$, we use (13) to find a lower bound for $N$ that gives $\|\mathbf{Y} - \mathbf{p}\|_\infty < \epsilon$ w.p. $1 - \beta$ when $\mathbf{Y}$ is sampled with $\rho$-tCDP. For simplicity, we consider a uniform prior: $\alpha_i = \alpha > 0$ for all $i$. Thus, $\rho = \frac{1}{2}\Delta_2^2 \psi'(\alpha - \gamma)$, where $\gamma$ might be chosen according to Corollary 2. We consider the two following regimes:

**High-privacy regime:** $\alpha - \gamma > 1$. In this case, $\psi'(\alpha - \gamma) \approx 1/(\alpha - \gamma)$. From $\rho = \frac{1}{2}\Delta_2^2 \psi'(\alpha - \gamma)$, we have $\alpha \approx \Delta_2^2/2\rho + \gamma$. Replacing $\alpha_0$ by $d\alpha$ in (13) yields the sample size:

$$N = \Omega\left(\frac{\log(2d/\beta)}{\epsilon^2} + \frac{d}{\epsilon}\left(\frac{\Delta_2^2}{2\rho} + \gamma\right)\right), \tag{14}$$

for the desired accuracy.

**Low-privacy regime:** $\alpha - \gamma < 1$. This is the same as above, except now we have $\psi'(\alpha - \gamma) \approx 1/(\alpha - \gamma)^2$, which implies $\alpha \approx \Delta_2/(2\rho)^{1/2} + \gamma$. The sample size that guarantees the desired accuracy is:

$$N = \Omega\left(\frac{\log(2d/\beta)}{\epsilon^2} + \frac{d}{\epsilon}\left(\frac{\Delta_2}{\sqrt{2\rho}} + \gamma\right)\right). \tag{15}$$

Let us compare this result to the Gaussian mechanism, which adds a noise $\mathbf{Z} \sim N(0, \sigma^2 I_d)$ to the normalized histogram $\mathbf{p}$ directly. Thus the $\ell_2$-sensitivity in this case is $\Delta_2/N$. We have that the Gaussian mechanism is $\rho$-zCDP where $\rho = \frac{\Delta^2}{2N^2\sigma^2}$ [BS16]. Using the same argument as above, with probability at least $1 - \beta$, the following inequality holds for all $i$:

$$\|\mathbf{Z}\|_\infty \leq \sqrt{\frac{\log(2d/\beta)\Delta_2^2}{N^2\rho}}. \tag{16}$$

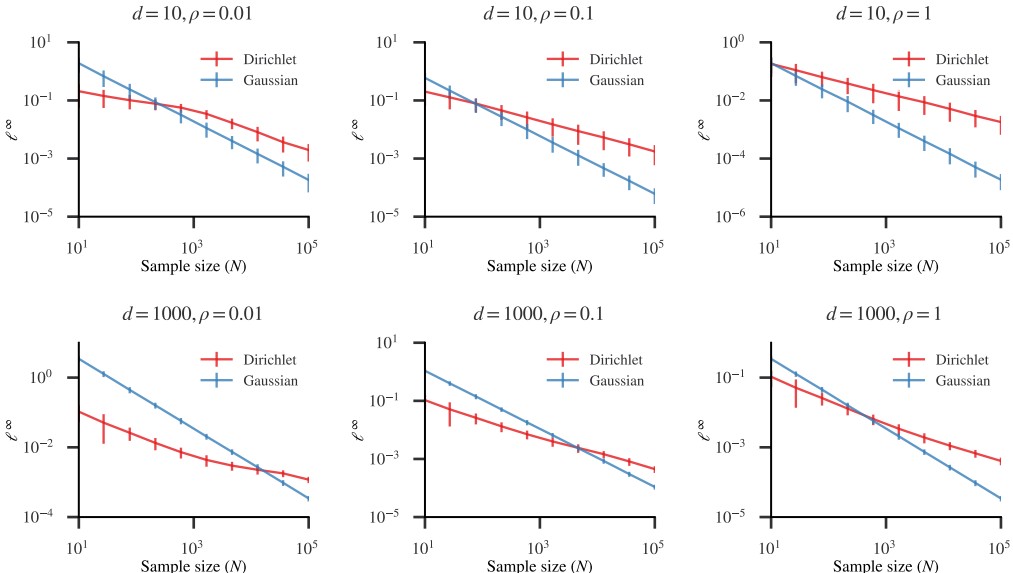

Figure 2: The $\ell^\infty$-accuracy, as a function of $N$, of Dirichlet posterior sampling ($\gamma = 1$) and Gaussian mechanisms for private normalized histograms ($\Delta_2^2 = 2$ and $\Delta_\infty = 1$). For each $N, d$ and $\rho$, we generated the inputs $\mathbf{x}_1, \ldots, \mathbf{x}_{200}$, where $\mathbf{x}_k \sim \mathrm{Multinomial}(\mathbf{q}_k)$ and $\mathbf{q}_k \sim \mathrm{Dirichlet}(5, \ldots, 5)$.

Hence, the sample size of $N = \Omega\left(\sqrt{\log(2d/\beta)\Delta_2^2/\rho\epsilon^2}\right)$ guarantees the desired accuracy. Comparing this to (14), if we assume $\epsilon < 1$, the AM-GM inequality tells us that

$$\frac{\log(2d/\beta)}{\epsilon^2} + \frac{d\Delta_2^2}{\rho\epsilon} > \frac{\log(2d/\beta)}{\epsilon^2} + \frac{\Delta_2^2}{\rho} \geq 2\sqrt{\frac{\log(2d/\beta)\Delta_2^2}{\rho\epsilon^2}}. \tag{17}$$

The inequality (17) implies that the Gaussian mechanism requires less sample than the Dirichlet mechanism in order to guarantee the same level of accuracy. The Gaussian mechanism is also better in the low-privacy regime as the $\rho$ in (15) satisfies $\sqrt{\rho} < \rho$ and $\Delta_2 \approx \Delta_2^2$, leading to the same inequality (17). Nonetheless, the decay in (16) is linear in $d$, while that in (13) has $\alpha_0 = d\alpha$ in the denominators. This observation suggests that, when $\mathbf{x}$ is a sparse histogram i.e. when $N \leq d$, the $\ell^\infty$-accuracy of the Dirichlet mechanism is smaller than that of the Gaussian mechanism. This conclusion is supported by our simulation in Figure 2. We see that the $\ell^\infty$-accuracy of the Dirichlet mechanism is smaller than that of the Gaussian mechanism for small $N$ when $d = 1000$. The code for all experiments in this study can be found in the supplemental material.

## Potential negative societal impacts

It is important to note that, when $\rho$ becomes unacceptably large (e.g., $\rho = 10^4$), the sampling is far away from being private. Thus any organization that deploys the posterior sampling on sensitive data must not vacuously refer to this study and claim that its algorithm is private. It is the organization's responsibility to fully publish the prior parameters, and educate its users/customers on differential privacy and how the privacy guarantees are calculated.

It is desirable that differentially private algorithms are accurate for the task at hand, especially when the data is used for important decision-making. Thus, one needs to make sure that there is enough sample to achieve the desired level of accuracy. For a large differentially private system, privacy budgets need to be allocated to the parts that require accurate outputs.

Lastly, one must be careful with the choice of prior parameters; if a uniform prior is used, smaller groups will suffer a relatively larger statistical bias. As a result, private statistics of small populations (such as ethnic or racial minorities) will be relatively less accurate. One way to get around this issue is to (privately) impose larger prior parameters on larger populations.

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
