# OpenReview forum: "Differential Privacy of Dirichlet Posterior Sampling"
_NeurIPS.cc/2021/Conference — NeurIPS 2021 Submitted_

### Official Review · Reviewer_Qa5w · 2021-07-12

**Rating:** 7
**Confidence:** 4

**Summary:**

This paper proposes a method for releasing differentially private samples from Dirichlet posterior distribution. Authors present a privacy analysis for their proposed method and demonstrate that the method out performs Gaussian mechanism under certain conditions. Besides the empirical evaluation, authors also derive analytical utility guarantees for the method.

**Limitations And Societal Impact:**

Yes

**Main Review:**

In this paper authors propose a method that draws a single sample from Dirichlet posterior distribution. The method is rather simple and draws from Dir$(x + \alpha)$, which is the posterior of multinomial observations $x$ given a Dirichlet prior Dir$(\alpha)$ for the multinomial probabilities. The privacy of the method is analysed with respect to the prior parameter $\alpha$ using truncanted CDP. Authors consider two use cases for this method (which I think are rather similar), and derive analytical expressions for the utility of these cases. Finally authors demonstrate that the proposed method outperforms Gaussian method in a histogram release task when the number of samples is small.

The analysis of the paper seems novel to me. I appreciate the authors' efforts for deriving analytical expressions between the privacy and utility. I can also see a use case for such method (for example the histogram release demonstrated in Sec 4.2), however I think the use cases might be rather limited. Thus I think the method might not be the most striking from usage perspective, but with the analysis it is a solid privacy paper.

Pros:
- (+++) Very clearly written paper
- (++) Novel privacy analysis in terms of Dirichlet prior parameter $\alpha$
- (++) Utility analysis in terms of sample size
- (+) Emprical results showing improvement over Gaussian mech.

Cons:
- (--) Use case might be limited
- (--) I'm not sure what is the actual difference between uses in 4.1 and 4.2

Minor comments/questions:
- Introduction, lines 30-32. Are you talking about drawing from posterior predictive? In that case I think Y should also be drawn for each sample.
- Theorem 1, proof. Is there really $\alpha'$ separate from $\alpha$ or is it a typo?
- Figure 2, I guess it would make more sense to call the y-axis $\ell_2$-error instead of $\ell_2$-accuracy. For me it seems odd to say that "We see that the $\ell_2$-accuracy of the Dirichlet mechanism is smaller than that of the Gaussian mechanism for small N when d = 1000.", when you actually want to point out that your method works better in the small N domain, i.e. it's more "accurate".


**Time Spent Reviewing:**

5

---

> ### Author Response · Authors · 2021-08-08
> **Response to Reviewer Qa5w**
>
> Thank you for the positive feedback. We address your comments below.
>
> >I'm not sure what is the actual difference between uses in 4.1 and 4.2
>
> In 4.1, we wish to sample from a Dirichlet posterior $\operatorname{Dirichlet}(\boldsymbol{x}+\boldsymbol{\alpha})$ (which may arise in Gibbs sampling or Thompson sampling). However, in most cases, the prior parameter $\boldsymbol{\alpha}$ is too small to satisfy a DP guarantees, so we have to enlarge them to $\boldsymbol{\alpha'}$ to obtain $\rho$-tCDP. To measure the "loss" incurred by enlarging the parameter, we compute the distance/divergence between two distributions $\operatorname{Dirichlet}(\boldsymbol{x}+\boldsymbol{\alpha})$ and $\operatorname{Dirichlet}(\boldsymbol{x}+\boldsymbol{\alpha'})$. Here, we chose the KL divergence as there is a closed form for the divergence between two Dirichlet distributions (given in Eq. (20) of Appendix B).
>
> In 4.2, we publish $\boldsymbol{Y}$ as a private release of an empirical distribution $\boldsymbol{p}=\boldsymbol{x}/N$. Thus an accurate value of $\boldsymbol{Y}$ is the one that is close to $\boldsymbol{p}$ (e.g. in $\ell^1$ or $\ell^{\infty}$ norm).
>
> >Introduction, lines 30-32. Are you talking about drawing from posterior predictive? In that case I think Y should also be drawn for each sample.
>
> The data synthesis approach in Lines 30-32 does not involve drawing from the posterior predictive; As the reviewer suggested, this involves drawing a large sample from the posterior distribution, which might lead to privacy violation. Instead, all of the studies that we cite in Line 33 synthesize data by drawing only a single point $\boldsymbol{Y}\sim \operatorname{Dirichlet(x_1+\alpha_1,\ldots,x_d+\alpha_d})$, then drawing a sample of an arbitrary size from $\operatorname{Multinomial}(\boldsymbol{Y})$.
>
> > Theorem 1, proof. Is there really $\boldsymbol{\alpha'}$ separate from $\boldsymbol{\alpha}$ or is it a typo?
>
> That is definitely a typo. Thank you for pointing it out.
>
> >Figure 2, I guess it would make more sense to call the y-axis $\ell^{\infty}$-error instead of $\ell^{\infty}$-accuracy. For me it seems odd to say that "We see that the $\ell^{\infty}$-accuracy of the Dirichlet mechanism is smaller than that of the Gaussian mechanism for small N when d = 1000.", when you actually want to point out that your method works better in the small N domain, i.e. it's more "accurate".
>
> We agree with the reviewer. A correct term should be  $\ell^{\infty}$-error.

---

> > ### Comment · Reviewer_Qa5w · 2021-08-31
> > **After the rebuttal**
> >
> > Thanks authors for addressing my concerns. I'm happy to keep my score as is.

---

### Official Review · Reviewer_QwTp · 2021-07-14

**Rating:** 3
**Confidence:** 4

**Summary:**


This paper studies differential privacy and the dirichlet mechanism.


**Ethical Concerns:**

--

**Limitations And Societal Impact:**


The text presents a comparison between the Guassian and Dirichlet mechanisms, and highlights around line 280 that the accuracy of the Dirichlet mechanism is better for spartse histograms and small N, but is silent about limitations, e.g., in the cases of dense hisograms and large N.

The paper doesn't discuss limitations, nor plans for future work where such limitations could be addressed.

**Main Review:**



The paper contains quite some phrases which aren't fully precise.  Often, a minor effort can make the text more precise without taking significantly more space.  In some other cases, the question is whether it is worthwhile to give technical details without giving the reader the context in which he can understand these details.  E.g.,
* Line 38: "DP allows us to quantify how much the privacy of the Dirichlet posterior sampling is affected by the prior parameters" -> what is meant here with "afect"?  Maybe you want to analyze how privacy guarantees depend on these parameters, but the cause of the change in privacy is in the publishing of the output so I wouldn't use "affect" for the parameters.
* Line 67: "any posterior sampling with the log-likelihood bounded by B is 4B-differentially private" -> which log-likelihood must be bounded by B ?  Do you mean classic \epsilon-differential privacy here (where you use B instead of \epsilon for some other reason) ?  Usually a loglikelihood B is a negative value, while \epsilon=4B must be positive to be able to consider \epsilon-DP.  Should the loglikelihood be upper bounded or lower bounded by B ?
* line 70: "if the condition on the log-likelihood is relaxed to the Lipschitz continuity with high probability," -> the uninformed reader reading only your summary may not understand how a log-likelihood bound |log(P(x))| < B (which is typically applied to random variables) can be relaxed to a Lipschitz continuity requirement (which is typically applied to functions).
* Line 74: "the sufficient statistics x" refers to some not-yet-introduced variables x on which "the" sufficient statistics is considered without explaining to the reader which sufficient statistics is being considered.
* Line 165: The theorem uses expressions involving x and x'.  x and x' are sample statistics of datasets, but it is unclear how they are sampled (e.g., with replacement / without replacement or other sample properties).  Even though we learn that x and x' are sample statistics from two adjacent datasets.
* Theorem 1, and also a few earlier sentences, use the term "one-time sampling", but this concept isn't defined (and also not a hit when provided to Google).  Do you mean "a single value x drawn from the distribution" (a single sample) or do you mean "at one time we draw many values x_1, x_2, x_3 ... without doing other things in between" (e.g., in contrast to MDP no decision can be made after a first sample to infleunce a later sample), or something else?

There are some minor language issue, often related to the use of articles where I wouldn't expect them or the absence of articles where I would expect them.  Also,
* Line 128: "... privacy" -> ... private

Some other comments:
* Line 142 saying "p(x|y)" suggests a conditional probability distribution where y is known and the distribution of drawing next x depends on y.  On the other hand, the equation between lines 144 and 145 suggests that we need to know x in order to draw Y.
* Line 150 discusses "Consider a special case where x = p", but p isn't introduced yet at that point: the only thing we know about p already is (from the notations section) that \sum_i p_i = 1.
* It turns out that p is an empirical distribution of a dataset.  The paragraph mentions the question how to make a private version of p (a problem which has been solved without using Dirichlet distributions), but doesn't immediately provide an answer.
* Figure 2: I guess that instead of "accuracy" (typically used in classification problems as the fraction of cases classified correctly, so ideally close to 1) you mean "norm" or "loss" (ideally small), at least this is what the Y-axis scale suggests.

The paper doesn't clearly define the problem it is trying to solve.  E.g., What values are sensitive and need to be protected from being revealed?  How does the "one-time sampling" differ from earlier work which considered the same problem but without "one-time sampling" ?


In conclusion, the paper has numerous minor issues (which individually can probably be easily solved but collectively may form a serious hurdle for a non-expert reader), introduces insufficiently clearly the problem (a problem formulation is the key part of a paper which should be understandable for all Neurips attendees), clarifies insufficiently in which way the contribution is novel and significant, and only in a limited way validates the work through experiments (with a single Gaussian baseline) where it would be possible to compare to other alternatives and discuss limitations and advantages.


**Time Spent Reviewing:**

3

---

> ### Author Response · Authors · 2021-08-08
> **Response to Reviewer QwTp - Part 2**
>
> >Line 142 saying "p(x|y)" suggests a conditional probability distribution where y is known and the distribution of drawing next x depends on y. On the other hand, the equation between lines 144 and 145 suggests that we need to know x in order to draw Y.
>
> As mentioned in Line 142, $p(\boldsymbol{x}\vert\boldsymbol{y})$ is the likelihood function, which is a function of the parameter $\boldsymbol{y}$ at a particular observed data $\boldsymbol{x}$. Here, we only use the likelihood function to derive the posterior density of $\boldsymbol{Y}$ through the Bayes' rule $p(\boldsymbol{y}\vert\boldsymbol{x}) \propto p(\boldsymbol{x}\vert\boldsymbol{y})p(\boldsymbol{y})$. Then we sample from the posterior. In short, we did not sample $\boldsymbol{x}$, but we did use the observed data $\boldsymbol{x}$ in order to draw $\boldsymbol{Y}$.
>
> >Line 150 discusses "Consider a special case where x = p", but p isn't introduced yet at that point: the only thing we know about p already is (from the notations section) that \sum_i p_i = 1.
>
> We recall that Line 150 writes "Consider a special case where $\boldsymbol{x}=\boldsymbol{p}$ is an empirical distribution...". This is a particular style of mathematical writing where "$\boldsymbol{x}=\boldsymbol{p}$ is an empirical distribution" means "$\boldsymbol{x}=\boldsymbol{p}$, where $\boldsymbol{p}$ is an empirical distribution". We might be more familiar with this style of writing in context of equation solving: "Let $x=x_0$ be a solution of $f(x)=0$".
>
> >It turns out that p is an empirical distribution of a dataset. The paragraph mentions the question how to make a private version of p (a problem which has been solved without using Dirichlet distributions), but doesn't immediately provide an answer.
>
> We publish $\boldsymbol{Y}$ as a private version of $\boldsymbol{p}$, as mentioned in Lines 150-151 that "...we want $\boldsymbol{Y}$ to be a private approximation of $\boldsymbol{p}$". Here, the definition of $\boldsymbol{Y}$, which is $\boldsymbol{Y}\sim \operatorname{Dirichlet}(r\boldsymbol{x}+\boldsymbol{\alpha})$, can be found in Lines 144-146.
>
> >Figure 2: I guess that instead of "accuracy" (typically used in classification problems as the fraction of cases classified correctly, so ideally close to 1) you mean "norm" or "loss" (ideally small), at least this is what the Y-axis scale suggests.
>
> A correct term should be either $\ell^{\infty}$-loss or $\ell^{\infty}$-error. Thank you for pointing it out.
>
> >The paper doesn't clearly define the problem it is trying to solve. E.g., What values are sensitive and need to be protected from being revealed? How does the "one-time sampling" differ from earlier work which considered the same problem but without "one-time sampling" ?
>
> Our problem is stated in the last two paragraphs of the introduction (Lines 34-44). In short, our goal is to calculate the differential privacy guarantees provided by the Dirichlet posterior sampling. By the definition of differential privacy, this means that the posterior sampling over *any types* of sensitive dataset (e.g. medical or financial data) can be used to protect any information about individuals in the dataset.
>
> Even though researchers have come up with many effective DP algorithms, there are still many randomized algorithms that have not been formally analyzed for their DP guarantees. Among those algorithms, we specifically focus on the Dirichlet posterior sampling, which is the main component of many Bayesian algorithms e.g. Gibbs sampling and Thompson sampling. Thus in contrast to earlier DP works which focus on finding new DP algorithms, our work provides a DP analysis of an algorithm that arises naturally in Bayesian analysis,
>
> Note that there are also closely related works that prove DP guarantees for posterior samplings with various types of conjugate pairs. See Section 1.2, where we list the differences between their results and ours.
>
> >The text presents a comparison between the Guassian and Dirichlet mechanisms, and highlights around line 280 that the accuracy of the Dirichlet mechanism is better for spartse histograms and small N, but is silent about limitations, e.g., in the cases of dense hisograms and large N.
>
> We point out that the analysis in Lines 268-277 assumes that $N$ has lower bounds (in high- and low-privacy regimes) according to Eq. (14) and (15), both of which implies that $N=\Omega(d\alpha/\epsilon)$, with $\epsilon<1$. Thus, the conclusion in Lines 274-277 that the Gaussian mechanism is preferable to the Dirichlet posterior sampling holds for large $N$ and dense histograms.
>
> >The paper doesn't discuss limitations, nor plans for future work where such limitations could be addressed.
>
> As pointed out in the Paper Checklist, we do in fact discuss (1) a situation under which the Gaussian mechanism is preferable to the Dirichlet posterior sampling in Lines 274-277, and (2) a limitation of the guaranteed upper bound of tCDP, with a potential solution, in the paragraph following to the proof of Theorem 1 (Lines 180-186).

---

> ### Author Response · Authors · 2021-08-08
> **Response to Reviewer QwTp - Part 1**
>
> Thank you for the feedback. We address your concerns below.
>
> >Line 38: "DP allows us to quantify how much the privacy of the Dirichlet posterior sampling is affected by the prior parameters" -> what is meant here with "afect"? Maybe you want to analyze how privacy guarantees depend on these parameters, but the cause of the change in privacy is in the publishing of the output so I wouldn't use "affect" for the parameters.
>
> Thank you for the correction. We tried not to overuse the word "privacy guarantees" too much in the introduction by replacing it by "privacy". However, as pointed out by the reviewer, these two words are not exchangeable. This will be fixed in the next version of the paper and we will be more careful in the future.
>
> >Line 67: "any posterior sampling with the log-likelihood bounded by B is 4B-differentially private" -> which log-likelihood must be bounded by B ? Do you mean classic \epsilon-differential privacy here (where you use B instead of \epsilon for some other reason) ? Usually a loglikelihood B is a negative value, while \epsilon=4B must be positive to be able to consider \epsilon-DP. Should the loglikelihood be upper bounded or lower bounded by B ?
>
> The term likelihood in "log-likelihood" refers to the likelihood function appears in the derivation of the posterior distribution: if $f(\theta|x)$ is the posterior distribution of $\theta$ after observing $x$, then $p(\theta|x)=\frac{p(x|\theta)p(\theta)}{p(x)}$. Here, $p(\theta)$ is the prior distribution, and $p(x|\theta)$ is the likelihood function.
>
> The term $4B$-differentially private here refers to $\epsilon$-differential privacy, where we set $\epsilon=4B$. Alternatively, we may write "any posterior sampling with the log-likelihood bounded by $\epsilon/4$ is $\epsilon$-differentially private". Since the meaning of both sentences are the same, it is up to the authors' preference which definition they use.
>
> For any function $f$, "$f$ is bounded" is a standard mathematical term which means that $f$ is *both* bounded above and bounded below. In particular, "$f$ is bounded by $B$" means that $f$ is bounded above by $B$ and bounded below by $-B$ (which automatically implies that $B\geq 0$). See for example page 73 in Apostol (1967).
>
> >line 70: "if the condition on the log-likelihood is relaxed to the Lipschitz continuity with high probability," -> the uninformed reader reading only your summary may not understand how a log-likelihood bound |log(P(x))| < B (which is typically applied to random variables) can be relaxed to a Lipschitz continuity requirement (which is typically applied to functions).
>
> The term "Lipschitz" of the log-likelihood $\ell(\boldsymbol{x},\theta)=\log p(\boldsymbol{x}|\theta)$ is with respect to $\boldsymbol{x}$, and the probability in "with high probability" is the prior probability of $\theta$.
>
> We will replace the statement:
> * "if the condition on the log-likelihood is relaxed to the Lipschitz continuity with high probability,"
>
> by
> * "if the log-likelihood $\ell(\boldsymbol{x},\theta)=\log p(\boldsymbol{x}|\theta)$ is Lipschitz continuous in $\boldsymbol{x}$ with high prior probability of $\theta$."
>
> in the next version of the paper.
>
> >Line 74: "the sufficient statistics x" refers to some not-yet-introduced variables x on which "the" sufficient statistics is considered without explaining to the reader which sufficient statistics is being considered.
>
> The confusion arose from using "*the* sufficient statistics $\boldsymbol{x}$" instead of "*a* sufficient statistics $\boldsymbol{x}$". To be more precise, $\boldsymbol{x}$ is a vector of any sufficient statistics of an exponential family.
>
> We will replace
> * "In the case that the sufficient statistics $\boldsymbol{x}$ has finite $\ell^1$-sensitivity,..."
>
> by
> * "Let $\boldsymbol{x}$ be a vector of any sufficient statistics of an exponential family. In the case that $\boldsymbol{x}$ has finite $\ell^1$-sensitivity,..."
>
> in the next version of the paper.
>
> >Line 165: The theorem uses expressions involving x and x'. x and x' are sample statistics of datasets, but it is unclear how they are sampled (e.g., with replacement / without replacement or other sample properties). Even though we learn that x and x' are sample statistics from two adjacent datasets.
>
> We point out that the statement in Theorem 1 contains "...sample statistics of any two datasets...". Here, "sample statistic" stands for any statistic that can be computed from the dataset e.g. mean, variance, counts, which can always be calculated even if the data did not come from any particular distribution. Thus there is no sampling involved in the definition of sample statistics. We can simply replace "sample statistics" by "outputs of an algorithm" but we would like to give this article more of a statistical flavor since it is mainly focused on Bayesian statistics.
>
> >Theorem 1, and also a few earlier sentences, use the term "one-time sampling", but this concept isn't defined (and also not a hit when provided to Google). Do you mean "a single value x drawn from the distribution" (a single sample) or do you mean "at one time we draw many values x_1, x_2, x_3 ... without doing other things in between" (e.g., in contrast to MDP no decision can be made after a first sample to infleunce a later sample), or something else?
>
> In our definition of one-time sampling, we draw *a single* observation from the probability distribution. As the reviewer pointed out, this term might cause confusion on whether we draw a single observation, or many observations at once, so we will add the definition of one-time sampling after its first appearance in the next version of the paper.
>
> References:
>
> Apostol, T.M. (1967) Calculus volume 1, 2nd edition, Wiley, New York.

---

> > ### Comment · Reviewer_QwTp · 2021-08-25
> > **comments on "Response to Reviewer QwTp - Part 1"**
> >
> > The authors propose several meaningful improvements / reformulations.
> > In general, I feel important recommendations remain to (a) include a systematic and clear(er) problem statement, and (b) use safe language which has a low risk of being ambiguous.
> >
> > Some further minor comments below:
> >
> > >>   Line 67: "any posterior sampling with the log-likelihood bounded by B is 4B-differentially private" -> which log-likelihood must be bounded by B ? Do you mean classic \epsilon-differential privacy here (where you use B instead of \epsilon for some other reason) ? Usually a loglikelihood B is a negative value, while \epsilon=4B must be positive to be able to consider \epsilon-DP. Should the loglikelihood be upper bounded or lower bounded by B ?
> >
> > > The term likelihood in "log-likelihood" refers to the likelihood function appears in the derivation of the posterior distribution: if
> > is the posterior distribution ...
> >
> > Indeed.  Still, it may help a non-expert reader to explain which posterior distribution you refer to.  More generally, in the domain of machine learning it is good practice to include a "problem statement" which describes what kind of prior knowledge one has, how data is generated, what is the model or variable one wants to learn/estimate, what is the objective function, what information exactly is sensitive, etc.  Doing so may take a few lines of text but a systematic and clear problem statement often avoids various opportunities for confusion later on.
> >
> > > "f is bounded" is a standard mathematical term which means that is both bounded above and bounded below. In particular, " is bounded by " means that is bounded above by and bounded below by (which automatically implies that ). See for example page 73 in Apostol (1967).
> >
> > This is quite possible.  Still, not all members of the Neurips public are perfect mathematicians.  I have seen less careful authors use "f is bounded" with different meanings in mind.  In fact, Neurips is not ICML or similar conference where you can assume all participants are from the domain of machine learning.  The Neurips website says: "The Thirty-Fifth Annual Conference on Neural Information Processing Systems (NeurIPS 2021) is an interdisciplinary conference that brings together researchers in machine learning, computational neuroscience, statistics, optimization, economics, computer vision, natural language processing, computational biology, and other fields.".
> >
> > In this case, even if the majority of mathematicians would agree with the implicit convention you use, it is very easy to avoid any possible risk at confusion by just saying "|f| is bounded" (rather than "f is bounded").  This language is safer and in this case doesn't use significantly more space.
> >
> > > Here, "sample statistic" stands for any statistic that can be computed from the dataset
> >
> > Yes, the confusion can come from the problem that "sample" often refers to the result of sampling, while in statistics "sample" is used as a synonym of a dataset.  "sample statistic" could hence refer both to "statistic computed on a dataset" and "statistic computed on a sample of the dataset (which in the field of statistics is sometimes also called subsample).  Maybe it is therefore safer to simply say "statistic of a dataset" or "dataset statistic" since such expression doesn't have such risk of ambiguity (and doesn't use more space).
> >
> > > where "x=p is an empirical distribution" means "x=p, where p is an empirical distribution".
> >
> > This explains the notation, but not fully how to obtain p.  "empirical distribution" suggests the (unique) distribution assigning to each instance in the dataset a probability equal to 1 divided by the dataset size, in which case I would expect "the empirical distribution" (rather than "an empirical distribution"), else if there are several ways to obtain p it may be clearer to be more explicit about what properties p should satisfy.

---

> > > ### Author Response · Authors · 2021-08-25
> > > **Response to Reviewer QwTp's comments**
> > >
> > > We thank the reviewer for the comments.
> > >
> > > >Indeed. Still, it may help a non-expert reader to explain which posterior distribution you refer to.
> > >
> > > We agree that some background materials in Bayesian inference might be helpful for non-expert readers. Due the page limit however, we were not able to go into details on both differential privacy and Bayesian inference. So we chose not to expand on the basic setup in Bayesian inference and instead focused on our specific case where the prior is a Dirichlet distribution and the likelihood is of the form $p(\boldsymbol{x}\mid\boldsymbol{y})\propto \prod y_i^{x_i}$ as stated in Section 2.3. As there is no mention on how the Dirichlet posterior is obtained from the likelihood and the prior, we will add a line that clarifies this in Section 2.3 in the next version of the paper.
> > >
> > > >More generally, in the domain of machine learning it is good practice to include a "problem statement" which describes what kind of prior knowledge one has, how data is generated, what is the model or variable one wants to learn/estimate, what is the objective function, what information exactly is sensitive, etc. Doing so may take a few lines of text but a systematic and clear problem statement often avoids various opportunities for confusion later on.
> > >
> > > Even though there is no dedicated section on a problem statement, we provide a section on overview of our results (Section 1.1) from which our problem statement can be implied. For example, it states that "We study the role of the prior parameters in the [differential] privacy of the Dirichlet posterior sampling." which implies that our problem is "how much the prior parameters affect the differential privacy guarantee of the Dirichlet posterior sampling."  Then, the precise statement can be obtained by combining this section with the Background (Section 2), which includes the definitions of differential privacy and the Dirichlet posterior sampling. Specifically, Section 2.3 tells us about the prior knowledge on inputs of the Dirichlet posterior sampling: "$x\in\mathbb{R}_{\geq 0}$ consists of sample statistics of the dataset.", Section 2.1 tells us about the privacy models. As it is mentioned in Section 1.1 that we prove a tCDP guarantee, the definition of tCDP (Definition 2.3) precisely tells us what "proving a tCDP guarantee" entails. With this, we think that adding a section of problem statement might be redundant considering the page limit. Nonetheless, for readers outside of differential privacy, we will mention right after Definition 2.1 that "differentially private algorithms aim to protect private details of each individual (a single row) contained in its input".
> > >
> > > >In this case, even if the majority of mathematicians would agree with the implicit convention you use, it is very easy to avoid any possible risk at confusion by just saying "|f| is bounded" (rather than "f is bounded"). This language is safer and in this case doesn't use significantly more space.
> > >
> > > We agree with the reviewer that some confusion may arise from this convention. We will simply change from "$f$ is bounded" to "$\rvert f \rvert$ is bounded" to avoid such confusion in the next version of the paper.
> > >
> > > >Maybe it is therefore safer to simply say "statistic of a dataset" or "dataset statistic" since such expression doesn't have such risk of ambiguity (and doesn't use more space).
> > >
> > > The reviewer made good suggestions; we will replace "...sample statistics of a dataset..." with simply "...statistics of a datasets..." in the next version of the paper.
> > >
> > > >This explains the notation, but not fully how to obtain p. "empirical distribution" suggests the (unique) distribution assigning to each instance in the dataset a probability equal to 1 divided by the dataset size, in which case I would expect "the empirical distribution" (rather than "an empirical distribution"), else if there are several ways to obtain p it may be clearer to be more explicit about what properties p should satisfy.
> > >
> > > We now understand the reviewer's original comment, and we are sorry for the confusion. By "empirical distribution" we refer to the one derived from a single attribute of the dataset (there are multiple empirical distributions since a dataset can have multiple attributes." We will replace "**an** empirical distribution derived from the dataset" with "**the** empirical distribution of an attribute of the dataset" in the next version of the paper.

---

### Official Review · Reviewer_ghaZ · 2021-07-17

**Rating:** 7
**Confidence:** 1

**Summary:**

The paper studies releasing a single sample from a Dirichlet posterior with several variants of differential privacy in the two specific tasks of Dirichlet sampling and private normalized histogram publishing. Both privacy and utility guarantees are analyzed by the authors. The authors compare the results with Gaussian mechanism theoretically and empirically using a simple simulation.

**Limitations And Societal Impact:**

The paper contains a section on "Potential negative societal impacts" which reasonably discussed the implications of the usage of the method in an industrial setting.

**Main Review:**

The paper is well-organized and easy to follow. The related works are summarized and allow the reader to compare the drawbacks of existing methods relatively to the proposed methods. All the notations and background definitions are clearly stated. While the paper contains a significant amount of theoretical derivations, many of them are accompanied by intuitions and interpretations (such as discussion of low/high privacy regimes or illustrations of  behavior of the computed quantities as a function of parameters, such as Fig. 1). I was not able to thoroughly verify all the derivations, but the math seems sounds. Overall, i believe the paper makes a good contribution and will be interesting to the researchers in the area.

**Time Spent Reviewing:**

4

---

### Official Review · Reviewer_g7yk · 2021-08-01

**Rating:** 6
**Confidence:** 4

**Summary:**

Summary: This paper gives a precise analysis of the privacy loss, as measured by differential privacy, from drawing samples from the Dirichlet posterior distribution on multinomial parameters conditioned on a sensitive data set.

**Limitations And Societal Impact:**

Well addressed in a separate section.

**Main Review:**

The paper has a single main result, namely a precise analysis of the distribution of the privacy loss—in the sense of differential privacy—of the algorithm which takes a sensitive data set as input and a nonsensitive Dirichlet prior on parameters of a multinomial distribution, constructs a posterior distribution on the parameters, and samples a single value from that distribution.

The main result (Theorem 1) is stated in terms of truncated concentrated DP, though I think it might be easier to state it in terms of Renyi DP (as a function of the $\alpha$ vector and the “order” $\lambda$) and then derive the tCDP and $(\epsilon,\delta)$-DP statements as corollaries. (I also think it would be worthwhile, in the corollaries, setting $\Delta_\infty$ and $\Delta_2$ to particular values, say 1, to get clear statements.)

I have very few comments on the writing: the paper is clear and easy to understand and, I think, accurately represents and discusses the contributions. The only question here is significance: the paper is providing updated calculations for a well-know idea. It definitely should be published in an archival and refereed venue. I’m less sure that it meets the bar for NeurIPS. If the committee ultimately decides not to accept the paper, I would encourage the authors to submit to an appropriate journal.


**Time Spent Reviewing:**

3

---

> ### Author Response · Authors · 2021-08-08
> **Response to Reviewer g7yk**
>
> Thank you for your review and valuable comments. Our responses are listed below.
>
> >The main result (Theorem 1) is stated in terms of truncated concentrated DP, though I think it might be easier to state it in terms of Renyi DP and then derive the tCDP and -DP statements as corollaries.
>
> This is a really good point. Restating the main result in terms of Renyi DP would definitely make it easier to digest the paper, especially when people are more familiar with Renyi DP than tCDP. Thus we will make the suggested change in the next version of the paper.
>
> >The only question here is significance: the paper is providing updated calculations for a well-know idea.
>
> In terms of significance, our result can be used for differential privacy analysis of Bayesian learning algorithms that involve Dirichlet posterior sampling, such as Gibbs sampling and Thompson sampling. In addition, we believe that the result will become useful when one wants to perform these Bayesian algorithms on noisy data with either Laplace or Gaussian noises; a standard DP analysis would just consider the Dirichlet posterior sampling as a post-processing step of the Laplace/Gaussian mechanism, but our result here suggests that the Dirichlet sampling can provide additional Renyi DP or $(\epsilon,\delta)$-DP guarantees to the algorithms (however, since the noise mechanisms destroy the global sensitivity of the data, it requires further analysis to calculate the DP guarantees of the post-noise sampling).
>
> We also would like to add that, even though the use of second-order Taylor approximation is not novel, our prove works out beautifully for the Dirichlet density as it can be approximately (i.e. modulo the normalizing factor) written as a product of functions of individual parameters. This might not be the case for any other multivariate distributions. For example, we can not apply the same proof to the multivariate normal, since the parameters are highly intertwined in the density function, unless the covariance matrix is diagonal.

---

### Decision · Program_Chairs · 2021-09-28

**Decision:**

Reject

**Comment:**

Reviewers generally appreciated the paper's clean problem formulation, algorithmic result, and analysis. There was consensus that the result is something that should be recorded in the literature. The two main downsides are 1) there is not much conceptual innovation behind the paper's algorithmic approach. Obtaining privacy by sampling once from an updated posterior distribution is a known idea, and the paper's novelty is in the analysis of privacy and utility. 2) Reviewer QwTp pointed out some editorial concerns that should be addressable in a revision of the paper. While the paper is technically sound and should be published, we did not find it exciting enough to meet the bar for NeurIPS.

Technical comment: The authors should justify why they measure utility using expected KL divergence with a particular ordering between P and Q. The way the paper is written, it seems this is just done for mathematical convenience rather than for relevance, e.g., to downstream inference tasks.

**Consistency Experiment:**

NeurIPS has a long history of experimentation. In 2014, NeurIPS ran an experiment in which 10% of submissions were reviewed by two independent committees to quantify the randomness in the review process. This year, we repeated a variant of this experiment to see how the quality of the review process has changed over time.  This paper was part of the experiment and was therefore assigned to two committees (consisting of reviewers, an Area Chair, and a Senior Area Chair) that reached independent decisions.  If both committees made the same recommendation, this recommendation was followed. If a single committee recommended acceptance, the paper was accepted (with the exception of a few cases in which the other committee identified what we considered a fatal flaw, e.g., an error in a key result).

Both committees reached the same decision: **Reject**

The other committee assigned to the paper recommended **Reject**.  You can find the other set of reviews, along with any follow up discussion with the authors here:
https://openreview.net/forum?id=Pk972F7J8Um